# A domain-knowledge modeling of hospital-acquired infection risk in Healthcare personnel from retrospective observational data: A case study for COVID-19

Phat K. Huynh[ID][1,2], Arveity R. Setty[3,4], Quan M. Tran[5], Om P. Yadav[6], Nita Yodo[2], Trung Q. Le[ID][1,2]*

1 Department of Industrial and Management Systems Engineering, University of South Florida, Tampa, FL, United States of America, 2 Department of Industrial and Manufacturing Engineering, North Dakota State University, Fargo, North Dakota, United States of America, 3 University of North Dakota, Fargo, North Dakota, United States of America, 4 Sanford Hospital, Fargo, North Dakota, United States of America, 5 Department of Biological Sciences, University of Notre Dame, Notre Dame, Indiana, United States of America, 6 Department of Industrial and Systems Engineering, North Carolina A&T State University, Greensboro, North Carolina, United States of America

* tqle@usf.edu

**Data Availability Statement:** The data and the description can be accessible via: https://github.com/PhatKHuynh/COVID-19-datasets.

## Abstract

### Introduction

Hospital-acquired infections of communicable viral diseases (CVDs) have been posing a tremendous challenge to healthcare workers globally. Healthcare personnel (HCP) is facing a consistent risk of viral infections, and subsequently higher rates of morbidity and mortality.

### Materials and methods

We proposed a domain-knowledge-driven infection risk model to quantify the individual HCP and the population-level risks. For individual-level risk estimation, a time-variant infection risk model is proposed to capture the transmission dynamics of CVDs. At the population-level, the infection risk is estimated using a Bayesian network model constructed from three feature sets, including individual-level factors, engineering control factors, and administrative control factors. For model validation, we investigated the case study of the Coronavirus disease, in which the individual-level and population-level infection risk models were applied. The data were collected from various sources such as COVID-19 transmission databases, health surveys/questionnaires from medical centers, U.S. Department of Labor databases, and cross-sectional studies.

### Results

Regarding the individual-level risk model, the variance-based sensitivity analysis indicated that the uncertainty in the estimated risk was attributed to two variables: the number of close contacts and the viral transmission probability. Next, the disease transmission probability was computed using a multivariate logistic regression applied for a cross-sectional HCP

**Funding:** This work was supported by National Institute of General Medical Sciences of the National Institutes of Health (NIH) under Award Number U54GM128729 and National Science Foundation under Award Number 2119691. The funders had no role in study design, data collection and analysis, decision to publish, or preparation of the manuscript.

data in the UK, with the 10-fold cross-validation accuracy of 78.23%. Combined with the previous result, we further validated the individual infection risk model by considering six occupations in the U.S. Department of Labor O*Net database. The occupation-specific risk evaluation suggested that the registered nurses, medical assistants, and respiratory therapists were the highest-risk occupations. For the population-level risk model validation, the infection risk in Texas and California was estimated, in which the infection risk in Texas was lower than that in California. This can be explained by California's higher patient load for each HCP per day and lower personal protective equipment (PPE) sufficiency level.

## Conclusion

The accurate estimation of infection risk at both individual level and population levels using our domain-knowledge-driven infection risk model will significantly enhance the PPE allocation, safety plans for HCP, and hospital staffing strategies.

## 1. Introduction

Nosocomial infections (i.e., hospital-acquired infections) of communicable viral diseases (CVDs) (e.g., influenza virus, hepatitis A virus, and rotavirus infections) have posed huge challenges to public health organizations. Healthcare personnel (HCP) experience the highest risk [1–3] because of the direct or indirect contact with infected patients and virus-contaminated surfaces. Subsequently, these HCP may become widespread the virus to non-infectious patients, coworkers, and their family members. Although there has been an increasing number of hospital outbreaks of CVDs over the last decade, current containment and preventive measures in hospital settings usually overlook asymptomatic individuals and "super spreader" events [4, 5]. Hence a quantitative estimation of the infection risk in HCP is critical to mitigate and subsequently prevent nosocomial infections in hospitals. Furthermore, a precise measure of HCP infection risk is also important to address the epidemiological issues in hospital settings and provide information for personal protective equipment (PPE) allocation, safety plans for HCP, and staffing strategies.

Modeling nosocomial HCP infections in hospitals has been based on mathematical models to qualitatively capture the dynamics of CVDs and the effects of different control measures [6, 7] at population level. One traditional model of disease spread is the compartmental SEIR (Susceptible-Exposed-Infected-Recovered) epidemiological model [8]. It divides a population into four different compartments or sub-groups (susceptible, exposed, infected, and recovered individuals) and employs deterministic ordinary differential equations to model the spread of a CVD. In the literature, there are many variants of this model (e.g., SIS, SIRD, MSIR, and MSEIR model). These models consider the population as homogenous without individual interactions (e.g., patients and HCP); therefore, they fail to capture the individual contact process and the effects of individual risk and protective factors [9]. In other words, only the patient flow dynamics are captured at the population level, but the individual-level information is neglected. To address the "homogenous population" assumption and capture better transmission dynamics, spatial epidemic models [10, 11] and metapopulation models [12, 13] have been introduced, which can be considered as "heterogeneous models". The spatial modeling approach incorporates the spatial information of disease occurrences and transmission behaviors to estimate of the probability of infection transmission from an infectious to a susceptible

individual at a certain distance. The metapopulation models are the integration of the compartmental models and spatial epidemic models, which allows us to simulate the behaviors of a large population given a well-defined spatial distribution. In this approach, the entire population is separated into two different subpopulations with coupled disease transmission dynamics based on the subpopulation characteristics, and each subpopulation is mixed homogeneously. In addition, network models [14, 15] have been proposed to model the contact network elaborately without having the assumption that all individuals interact with each other with a regular random pattern [16]. The contact network concept was founded by the mathematical graph theory, and a graph as two fundamental components: vertices and edges. In the context of epidemiological modeling, a vertex can represent an individual, a subpopulation, or even an entire country, an edge is the link between vertices, which captures the disease transmissions and individual interactions. Despite the tremendous efforts in advancing the population-based mathematical epidemiological model, the advanced models, such as "heterogeneous models" and network models, are still limited in terms of modeling and characterizing the individuals' movements and interactions.

To overcome the limitations of the population-based models, individual-based models have been proposed to fully capture the complexity of individual behaviors and interactions with other individuals. First, complex systems approaches using cellular automata (CA) theory have been proposed to model location-specific dynamics of susceptible populations and the probabilistic nature of disease transmission [17, 18], which takes into account the individual movements of the populations. However, The major drawback of CA models is their insufficiency in characterizing the spatial-temporal information of individuals' movements and interactions [19]. Agent-based modeling (ABM) was proposed to address the limitations of CA models by accounting for the individual-level movement of disease carriers and the contact network of people [20]. Although the ABM approach can capture the spread of a CVD in a spatial region (e.g., hospital) over time and estimate the risk of viral infection, it requires a large amount of information of individuals' movement and high computational cost. Moreover, individuals' movements are highly restricted in hospital settings, especially for patients who have positive test results for infectious diseases.

Statistical models have been used as an alternative to mathematical models to quantify the population-averaged effects of protective or risk factors on the time-variant infection risk of HCP. Statistical models capture the disease transmission dynamics within the hospital, HCP-related risk factors of infection, and other patients and HCP as sources of infection [21]. Two classes of statistical models, namely measure of association and statistical survival analysis, have been proposed to estimate HCP infection risk. The measure of association approaches quantifies the relationship between the exposed and diseased HCP groups by using the adjusted odds ratio (aOR), risk difference (RD), and relative risk (RR) as the risk measures [2, 22–25]. To capture the changes of HCP's characteristics and infection risk over time, survival analysis models are used to estimate the HCP infection risk and the expected duration of time until a viral infection occurs [26, 27]. Although time-dependent variables have been considered in the survival analysis models, the stochastic nature of epidemiological dynamics and individual interactions have not been investigated.

To overcome the above research gaps, this paper proposes a probabilistic domain-knowledge-driven model of the infection risk of CVDs for HCP, which is a hybrid multi-scale approach that combines mechanistic modeling and statistical modeling approaches. The term "domain-knowledge-driven model" [28] refers to a class of statistical or machine learning models that leverage the expert knowledges and embed them into the models to enhance the performance, understanding, and validity. The domain knowledge we incorporated in our proposed probabilistic framework includes modes of CVD transmissions, significant risk and

protective factors of the infection risk, and disease transmission dynamics through patient compartments. The proposed model was formulated for the infection risk estimation at both individual and population levels with respect to three modes of transmissions: 1) direct contact of susceptible HCP with other infectious individuals including patients and coworkers, 2) airborne viruses, and 3) contaminated equipment and surfaces. The individual-level risk model was built based on the population grouping in the SEIR model with the consideration of the time-varying confounders to capture the dynamical contagious disease transmission mechanism. At the population-level, three subsets of features, which are introduced in Sub-section 2.2, were constructed and represented by a Bayesian network [29], from which the probability of transmission from patients to HCP was estimated. The main contributions of this paper are 1) a novel probabilistic model characterizes the dynamics of the disease transmission in HCP over time and 2) a domain-knowledge-driven risk analysis model that quantifies both the individual-level and population-level infection risk. The remainder of the manuscript is organized as follows: Section 2 discusses the model formulation and model validation; the results with sensitivity analysis and the case study on the COVID-19 are presented in Section 3, discussion and conclusions are provided in Sections 4 and 5.

## 2. Materials and methods

The proposed framework consists of two sub-models: (1) an individual-level infection risk model that quantifies the risk of infection of an HCP, and (2) a population-level model that estimates the infection risk under working conditions at a medical facility. The output from the first sub-model serves as an input for the estimation of the population infection risk in the second model. Other inputs, such as engineering control and administrative factors, were also considered in the estimation of population risk.

### 2.1. Individual-level infection risk model

The individual infection risk model aims to quantify the potential risk of infection associated with a healthcare worker subject to nosocomial infection, whose job functions require working in proximity of patients. The proposed individual-level infection risk model is formulated using the population grouping approach in the compartmental SEIR model [8], in which the population is divided into different compartments (i.e., Susceptible ($S$), Exposed ($E$), Infectious ($I$), or Recovered ($R$)). However, susceptible ($S$) and recovered individuals ($R$) cannot transmit the virus during the length of a hospital stay [8], hence we do not consider these compartments in our model. Moreover, we do not assume that the recovered patients confer immunity to reinfection when being released from isolation. HCP coworkers have also been shown to contribute significantly to virus spread within the healthcare setting if contracting a virus [26, 30]. To capture the virus transmission mechanism, the healthcare worker group ($HW$) is added to model the HCP-HCP transmission, and the infectious individuals are further classified into two sub-groups: the infection-confirmed group ($IC$) and the infection-suspected ($IS$) group. Infection-confirmed individuals are those who have lab-confirmed infections (*e.g.*, individuals have tested positive for COVID-19 using the polymerase chain reaction (PCR) test), and the infection-suspected group includes individuals who are suspected to have the virus infection because they developed symptoms but have never tested for the infectious disease. In total, four groups ($E$, $IC$, $IS$, $HW$) are considered to model the individual HCP infection risk. We denote the potential infection risk of the HCP $j$ at location $i$ (e.g., hospitals) over time from $t_1$ to $t_2$ as $PIR_{i,j}^{(t_1:t_2)}$, which is the cumulative risk of viral infection after contacting patients and contaminated surfaces. We denote $N_{E,j}^{(t_1:t_2)}$, $N_{IC,j}^{(t_1:t_2)}$, $N_{IS,j}^{(t_1:t_2)}$, and $N_{HW,j}^{(t_1:t_2)}$ as the number of exposed

cases, infection-confirmed, infection-suspected, and colleagues that an HCP $j$ has contacted with over the time $(t_1:t_2)$, which is denoted as $(\cdot)$ (e.g., $N_{E,j}^{(t_1:t_2)} = N_{E,j}^{(\cdot)}$).

An HCP $j$ is assumed to have $CC_{X,k}^{(t_1:t_2)}$ independent close contacts with an individual $k$. Next, we denote $p_{X,k \to j}^{(\cdot)}$ as the probability of viral transmission from individual $k$ to the HCP $j$, with $X \in \{E, IC, IS, HW\}$ being the compartment indicator of person $k$. Here, if the probability $p_{X,k \to j}^{(\cdot)}$ is constant, the viral transmission mechanism is modelled as a binomial process $Bin(CC_{X,k}^{(\cdot)}, p_{X,k \to j}^{(\cdot)})$ [31], and there are $N_{E,j}^{(\cdot)} + N_{IC,j}^{(\cdot)} + N_{IS,j}^{(\cdot)} + N_{HW,j}^{(\cdot)}$ binomial processes in total. The sequence of contacts of HCP $j$ ordered by time will be superscripted by person index $k(m)$ and compartment index $X(m)$ as follows:

$$\boldsymbol{C}^{(t_1:t_2)} = \{C_m^{k(m),X(m)} | k(m) = 1, \ldots, N_{X(m),j}^{(\cdot)}\} \tag{1}$$

where $X(m) = \{E, IC, IS, HW\}$, $m$ is the temporal order of close contacts from which the HCP $j$ contracts the virus, $C_m^{k(m),X(m)} = 1$ if the HCP $j$ contracts the virus at the $m^{th}$ close contact, $C_m^{k(m),X(m)} = 0$ otherwise. As a result, the risk $PIR_{i,j}^{(\cdot)}$, is estimated as:

$$PIR_{i,j}^{(\cdot)} = \sum_{m=1}^{|\boldsymbol{C}^{(\cdot)}|} P(C_m^{k(m),X(m)} = 1, \boldsymbol{C}_{1:m-1}^{k(m),X(m)} = \boldsymbol{0}) \tag{2}$$

where $|\boldsymbol{C}^{(\cdot)}|$ is the total number of contacts and $\boldsymbol{C}_{1:m-1}^{k(m),X(m)} = 0$ means all previous $m-1$ contacts are the failed transmissions. Given the assumption of independent close contacts, Eq (2) can be expressed as:

$$PIR_{i,j}^{(\cdot)} = \sum_{m=1}^{|\boldsymbol{C}^{(\cdot)}|} \left[ \prod_{r=1}^{m-1} (1 - p_{X(r),k(r) \to j}^{(\cdot)}) \right] p_{X(m),k(m) \to j}^{(\cdot)} \tag{3}$$

The expectation and variance of $PIR_{i,j}^{(\cdot)}$ are further investigated and presented in the S1 File. If we denote $TP_-^{j,k}$ and $TP_+^{j,k}$ as the patient admission time and the recovery time of an individual $k$ with whom the HCP $j$ has close contacts, the time interval $[TP_-^{j,k}, TP_+^{j,k}]$ is the virus exposure period for the HCP $j$ with the person $k$. Therefore, $p_{X(r),k(r) \to j}^{(t_1:t_2)}$ can be reduced to $p_{X(r),k(r) \to j}^{(\max\{t_1, TP_-^{j,k(r)}\}:\min\{t_2, TP_+^{j,k(r)}\})}$. If $p_{X,k \to j}^{(\cdot)}$ is time-invariant, a logistic regression model is established to estimate the probability $p_{X(r),k(r) \to j}^{(\cdot)}$ as:

$$\log \left[ \frac{p_{X(r),k(r) \to j}^{(\cdot)}}{1 - p_{X(r),k(r) \to j}^{(\cdot)}} \right] = \boldsymbol{Z}^T \boldsymbol{\beta} \tag{4}$$

$$p_{X(r),k(r) \to j}^{(\cdot)} = P\left( Y_{X(r),k(r) \to j}^{(\cdot)} = 1 \right) = \frac{\exp(\boldsymbol{Z}^T \boldsymbol{\beta})}{1 + \exp(\boldsymbol{Z}^T \boldsymbol{\beta})} \tag{5}$$

where $Y_{X(r),k(r) \to j}^{(\cdot)}$ is the indicator variable ($Y_{X(r),k(r) \to j}^{(\cdot)} = 1$ means that HCP $j$ has contracted the virus via the contact with person $k(r)$ and $Y_{X(r),k(r) \to j}^{(\cdot)} = 0$ if HCP $j$ has failed to contract the virus), $\boldsymbol{Z}$ is the covariate vector including the factors influencing the response and $\boldsymbol{\beta}$ is the coefficient vector. If $p_{X,k \to j}^{(\cdot)}$ varies over time, the constant $p_{X,k \to j}^{(\cdot)}$ assumption is relaxed by considering the cumulative distribution function that describes the probability of infection up to time $t$: $F(t) = P(T \le t) = 1 - \exp(- \int_0^t h(t)dt)$, in which $T$ is the infection time and $h(t)$ is the

hazard function. We assume $h(t) = 0$ over the time of no close contacts. Hence, $PIR_{i,j}^{(\cdot)} = P(t_1 \leq T \leq t_2)$ is:

$$PIR_{i,j}^{(\cdot)} = 1 - \exp\left(-\int_0^{t_2} h(t)dt\right) - \exp\left(-\int_0^{t_1} h(t)dt\right) = \sum_{m=1}^{|\boldsymbol{C}^{(\cdot)}|}\left\{1 - \exp\left(-\int_0^{\tau_m} h_m(t)dt\right)\right\} \quad (6)$$

$$h(t) = \begin{cases} 0 \; if \; t \notin [\tau_m, \tau'_m] \\ h_m(t) \; if \; t \in [\tau_m, \tau'_m] \end{cases}, \quad \forall m = 1, \ldots, |\boldsymbol{C}^{(\cdot)}| \quad (7)$$

where $[\tau_m, \tau'_m]$ is the time period of the $m^{th}$ close contact with person $k(m)$, and $h_m(t)$ is the cumulative infection time distribution function for the $m^{th}$ close contact. The probability $p_{X(r),k(r)\to j}^{(\cdot)}$ and $h_m(t)$ depend on various factors including HCP- dependent features, patient-dependent features, patient-HCP interactions, HCP-HCP interactions, and healthcare facilities' conditions.

## 2.2. Population risk indicator model

The population risk indicator quantifies the potential viral infection risk associated with a hospital/clinic over the time period $[t_1:t_2]$. The population risk, annotated as $PIR_i^{(t_1:t_2)}$, is interpreted as the probability that an HCP contracts the disease under working conditions at place $i$ given the information about the individual-level infection risk of all HCP at place $i$ and the external factors. At this level, external factors from engineering and administrative controls within the hospital are considered. Those are the factors that affect the population-level infection risk apart from the individual-level risk. Representative examples of engineering controls are high-efficiency air, ventilation rates at the workplace, and infection isolation rooms for aerosol generating procedures. Administrative controls include formal HCP training regarding PPE availability level, training on risk factors and resources to promote personal hygiene. The $PIR_i^{(t_1:t_2)}$ is computed using logistic function as:

$$PIR_i^{(\cdot)} = \left\{1 + \exp\left[-\sum_j \frac{f(\boldsymbol{PIR}_{i,j=1,\ldots n_{HCP}}^{(\cdot)}, \boldsymbol{F})}{\tau}\right]\right\}^{-1} \quad (8)$$

where $\boldsymbol{PIR}_{i,j=1,\ldots n_{HCP}}^{(\cdot)} = [PIR_{i,1}^{(\cdot)}, PIR_{i,2}^{(\cdot)}, \ldots PIR_{i,n_{HCP}}^{(\cdot)}]^T$ is the vector of individual infection risk estimates of a total number of $n_{HCP}$ HCP, $\tau$ is the scaling parameter, $\boldsymbol{F} = \{F_i\}$ is the vector of engineering control and administrative control factors. We denote $f(\cdot)$ as the abbreviated notation for the function of $PIR_{i,j}^{(\cdot)}$ and $\boldsymbol{F}$. When the working restriction policy is applied to a certain HCP, which forces him/her to be self-isolated at home, his/her individual risk will not be considered in Eq (8). The function $f(\cdot)$ can be simply formulated as a linear regression model such that:

$$f(\cdot) = \boldsymbol{\alpha}\boldsymbol{PIR}_{i,j=1,\ldots n_{HCP}}^{(\cdot)} + w_1 F_1 + \cdots + w_n F_n + b \quad (9)$$

where $\boldsymbol{\alpha}$, $w_i$, and $b$ are the model parameters. Alternatively, the population risk $PIR_i^{(\cdot)}$ is estimated using a Bayesian network when we have access to the domain knowledge that describes the relationships between the control factors and the infection risk at the population level and individual level. Here, the Bayesian network model [32] is employed to incorporate the domain knowledge that influences the virus spread. The network is formulated based on three subsets of factors from the literature that affect the risk of infection including 1) individual-

level factors, 2) engineering control factors, and 3) administrative control factors (see Fig 1). Individual-level factors include patient characteristics (e.g., time from exposure to symptom onset, clinical severity of patients), HCP-dependent factors (e.g., PPE sufficiency level, close contacts with patients, exposure level to infection, working hours per week), and intervention-related risks (e.g., endotracheal intubation, high flow nasal cannula (HFNC)). External factors include engineering control factors (e.g., ventilation rates, airborne infection isolation rooms) and administrative control factors (e.g., formal HCP training on PPE and disease risk factors, resources to promote personal hygiene). These factors are annotated as **ILF**, **ECF**, and **ACF**, respectively. Hence, using the chain rule of the Bayesian network [33], the risk $PIR_i^{(\cdot)}$ is

$$PIR_i^{(\cdot)} = P(X_{PIR_i^{(\cdot)}} = 1 | PIR_{i,j}^{(\cdot)}, \mathbf{ECF}, \mathbf{ACF}, \mathbf{ILF})$$

$$= \frac{P(X_{PIR_i^{(\cdot)}}, PIR_{i,j}^{(\cdot)}, \mathbf{ECF}, \mathbf{ACF}, \mathbf{ILF})}{P(\mathbf{ECF})P(\mathbf{ACF})P(\mathbf{ILF}|\mathbf{ACF}, \mathbf{ECF})P(PIR_{i,j}^{(\cdot)}|\mathbf{ILF})} \quad (10)$$

where $P(\cdot)$ is the probability function, and $X_{PIR_i^{(\cdot)}}$ is the indicator variable ($X_{PIR_{i,j}^{(\cdot)}} = 1$ indicates that an HCP contracts the disease and $X_{PIR_{i,j}^{(\cdot)}} = 0$ if they do not).

## 3. Results and covid-19 case study

### 3.1. Sensitivity analysis using simulated data

The variance-based sensitivity analysis was utilized to study the uncertainty of HCP's potential infection risk output caused by the variance of the input variables.

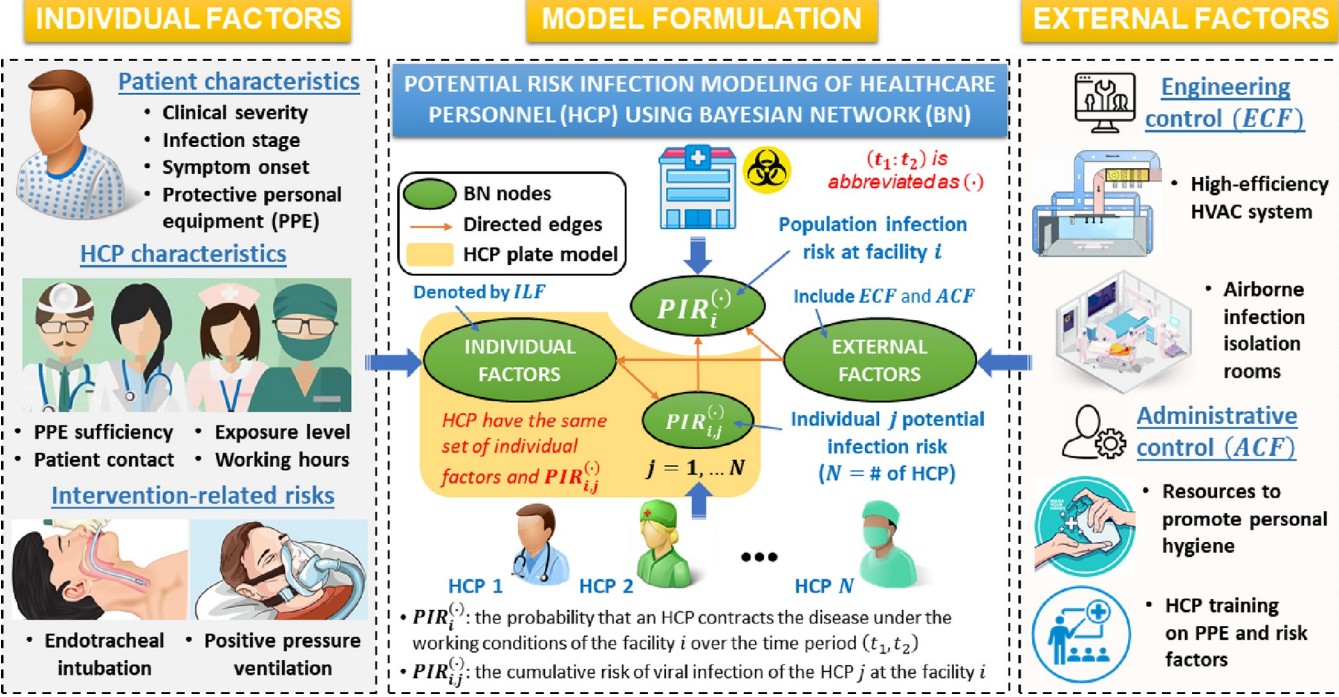

**Fig 1. Illustration of the contributions of the individual factors and external factors to the estimation of the infection risk of HCP in our model formulation.** The infection risk at both individual level and population levels can be estimated based on a Bayesian network formulation which has 4 main nodes, namely the individual-level risk $PIR_{i,j}^{(\cdot)}$, the population-level risk $PIR_i^{(\cdot)}$, the individual-level factors, and the external factors. The individual-level factors (**ILF**) include patient characteristics, HCP characteristics, and intervention-related risks, whereas the external factors consist of engineering control factors (**ECF**) and administrative control factors (**ACF**).

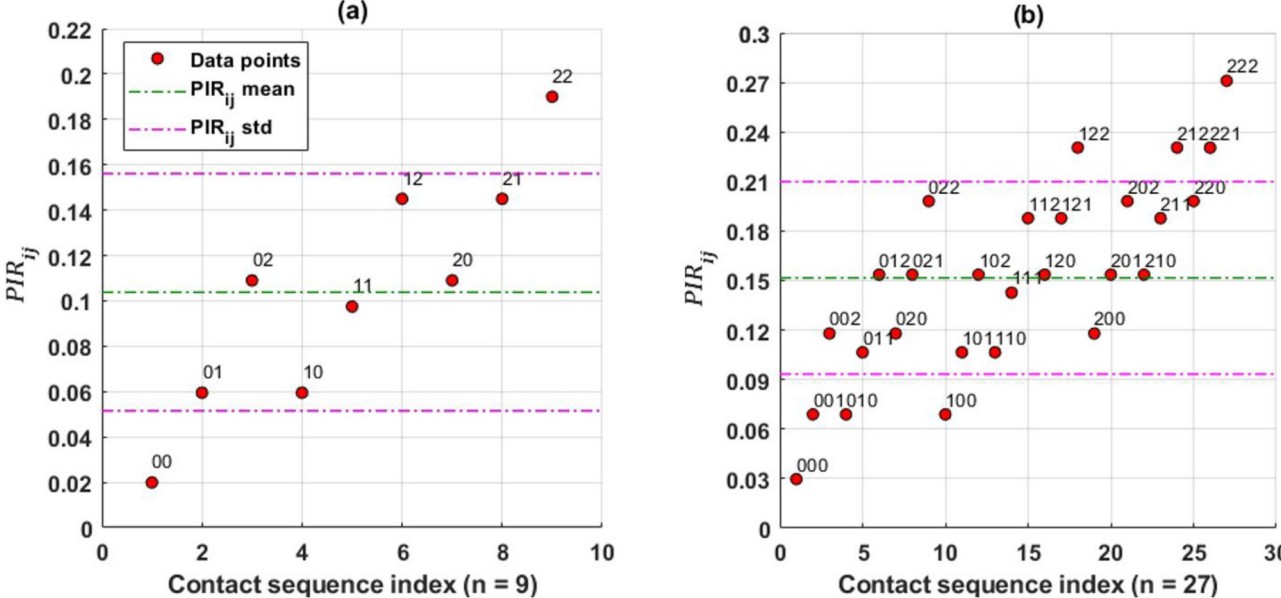

**Fig 2. Sensitivity analysis of the impact of probability of viral transmission and the number of close contacts on $PIR_{i,j}^{(t_1:t_2)}$.** We estimated values of $PIR_{i,j}^{(t_1:t_2)}$ for the synthesized data with three levels of $p_{X(m),k(m)\to j}^{(\cdot)}$: $P_{low} = 0.01$, $P_{medium} = 0.05$, $P_{high} = 0.1$. Panel (a): the estimated $PIR_{i,j}^{(t_1:t_2)}$ for $|C^{(\cdot)}| = 2$, i.e., two close contacts; therefore, there are $n = 3^2 = 9$ possible contact sequences with different combinations of $p_{X(m),k(m)\to j}^{(\cdot)}$ levels, and those combinations are encoded in the form $X_1 X_2 \ldots X_n$, where $X_1, X_2, \ldots, X_n \in \{0,1,2\}$, which corresponds to low, medium, and high levels of $p_{X(m),k(m)\to j}^{(\cdot)}$. The mean level of $PIR_{i,j}^{(t_1:t_2)}$ (green dash-dotted line) associated with its standard deviation indicated by purple dash-dotted lines are also plotted. Panel (b): the results for $|C^{(\cdot)}| = 3$ with $n = 3^3 = 27$ possible contact sequences.

**3.1.1. The measure of sensitivity of $PIR_{i,j}^{(\cdot)}$ to $p_{X(m),k(m)\to j}^{(\cdot)}$ and close contact sequence.** The dependence of the infection risk on the probability of viral transmission and close contact sequence for an HCP was analyzed. The $PIR_{i,j}^{(\cdot)}$'s for different numbers of close contacts $|C^{(\cdot)}|$ were estimated by Eq (3). For illustration, the results for $|C^{(\cdot)}| = 2$ and $|C^{(\cdot)}| = 3$ are shown in Fig 2.

According to the results, the mean level (± standard deviation) of $PIR_{i,j}^{(\cdot)}$ for $|C^{(\cdot)}| = 2$ was 0.1038±0.0523, which was lower than that for $|C^{(\cdot)}| = 3$ at 0.1516±0.0583. The mean value of the individual risk escalated together with the standard deviation values as the number of contacts increased. In addition, the estimated $PIR_{i,j}^{(\cdot)}$ was not influenced by the time order of the close contacts, e.g., the same $PIR_{i,j}^{(\cdot)} = 0.1065$ for three sequences: 011, 101, 110, where 0 and 1 are the encoded values for $P(Low)$ and $P(medium)$ respectively. The results are from the assumption of temporal independence between close contacts However, the risk would increase when the probability $p_{X(m),k(m)\to j}^{(\cdot)}$ for each contact raised to a higher value, hence the probabilities collectively contributed to the value of risk.

**3.1.2. Response surface of the mean and variance of $PIR_{i,j}^{(t_1:t_2)}$.** The measure of sensitivity of potential infection risk $PIR_{i,j}^{(t_1:t_2)}$ of the HCP $j$ at the place $i$ over time $(t_1: t_2)$ was investigated. We denote the mean level and the variance of $PIR_{i,j}^{(t_1:t_2)}$ of all sequences given the number of close contacts $|C^{(\cdot)}|$ as $E[PIR_{i,j}^{(\cdot)}]$ and $Var[PIR_{i,j}^{(\cdot)}]$, respectively. Next, we defined two levels of $p_{X(m),k(m)\to j}^{(\cdot)}$: $P_{low} \in (0,0.5)$ and $P_{high} = P_{low} + 0.3$, and derived the response surfaces of the $E[PIR_{i,j}^{(\cdot)}]$ and $Var[PIR_{i,j}^{(\cdot)}]$ with respect to two inputs $P_{low}$ and $|C^{(\cdot)}|$. As shown in Fig 3, the response

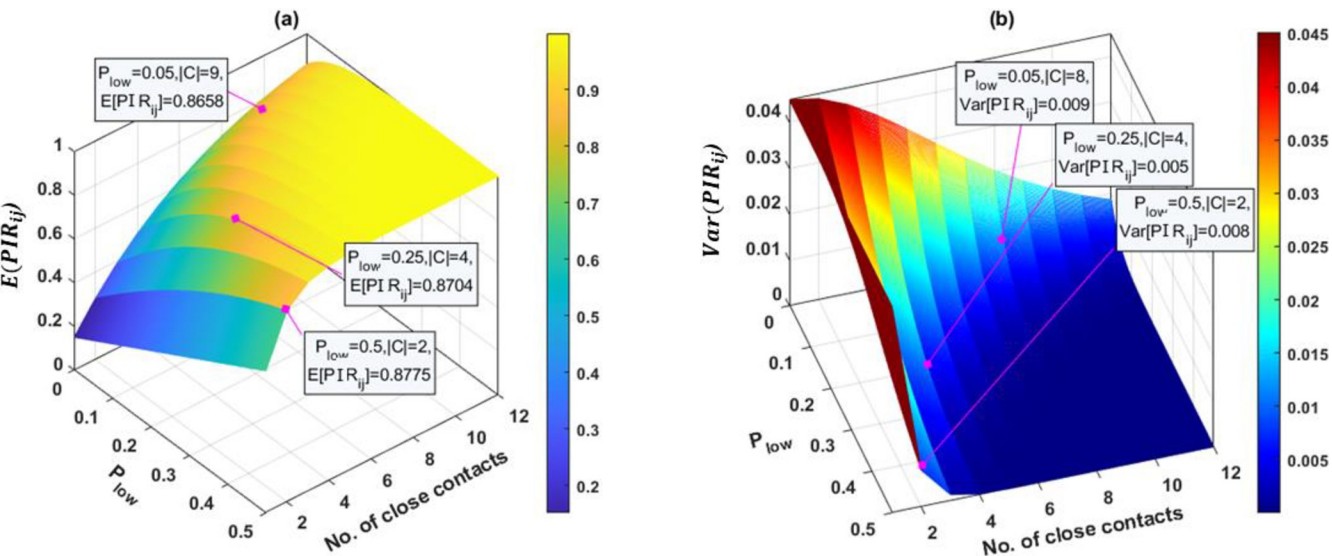

**Fig 3. Response surfaces of $E[PIR_{i,j}^{(t_1:t_2)}]$ and $Var[PIR_{i,j}^{(t_1:t_2)}]$ with respect to two input variables: Viral transmission probability and number of close contacts.** (a): the response surface of $E[PIR_{i,j}^{(t_1:t_2)}]$ subject to the change of $P_{low}$ and total number of close contacts $|C^{(\cdot)}| \in [1,12]$. A data set was synthesized with two levels of $p_{X(m),k(m)\to j}^{(\cdot)}$: $P_{low} \in (0,0.5]$ and $P_{high} = P_{low} + 0.3$. where the expectation $E[PIR_{i,j}^{(t_1:t_2)}]$ is the mean level of $PIR_{i,j}^{(\cdot)}$ of all possible contact sequences $C^{(\cdot)}$, which are the combinations of $P_{low}$ and $P_{high}$ in the sequence of length $|C^{(\cdot)}|$. Data tips at 3 values of $P_{low}$: 0.05, 0.2, 0.5 were created to indicate the cut-off values of $|C^{(\cdot)}|$ when $E[PIR_{i,j}^{(t_1:t_2)}]$ was significantly high. Similarly, (b) shows the response surface of $Var[PIR_{i,j}^{(t_1:t_2)}]$ of all possible sequences subject to the change of $P_{low}$ and $|C^{(\cdot)}|$. Three data tips at $P_{low} = \{0.05, 0.2, 0.5\}$ were included to show the threshold of $|C^{(\cdot)}|$ at which $Var[PIR_{i,j}^{(t_1:t_2)}]$ was sufficiently low.

surface of $E[PIR_{i,j}^{(\cdot)}]$ showed that a high probability of successful viral transmission $p_{X(m),k(m)\to j}^{(\cdot)}$ will result in an extremely high value of $E[PIR_{i,j}^{(\cdot)}]$, e.g., $E[PIR_{i,j}^{(\cdot)}] = 0.8336$ when $|C^{(\cdot)}|$ is only 3, $P_{low} = 0.3$, and $P_{high} = 0.7$.

### 3.2. Model validation using COVID-19 case study

**3.2.1. Case study description.** Data sets of HCPs with COVID-19 were used to validate the proposed model. Access to these data sources can be provided per requests or via the cited references. The validation was performed on three main components: the viral transmission probability model, the individual-level infection risk model, and the population-level risk model. The HCP's occupational infection risk to COVID-19, interim guidance regarding risk assessment and universal PPE policy issued by the CDC [41], and the risk factors for severe acute respiratory syndrome coronavirus (SARS-CoV-2) transmission in hospital settings from previous studies were also included to develop the model for the case study.

The major factors resulting in high risk for HCPs are 1) exposure to COVID-19 patients without using appropriate PPE, 2) involvement in aerosol-generating procedures and the interventions performed by physicians or nurses, and 3) contact with patients and colleagues during the incubation period. Many studies suggested that there is a significant association between PPE use and infection risk and that masks are the most consistent contributing measure to reduce the risk [34, 35]. A similar association was observed for other PPE, such as gowns, gloves, and eye protection. Other exposures and treatment practices (e.g., intubation involvement, patient care, or having contact with secretions) were found to link with increased infection risk for HCPs [36, 37]. Finally, given the implementation of a universal PPE policy, the high risk of infection among HCP also arises from contacting asymptomatic patients and colleagues who are in the early phase of viral infections [24].

Different regression models, including logistic regression, log-binomial, and Poisson, were used with the defined risk measures to estimate the COVID-19 infection risk among HCP groups [23–25, 38–45]. Statistical survival analysis models were also used to estimate the HCP's risk of contracting SARS COV-2 viruses and the expected duration of time until viral infection occurs. Shah et al. [27] modeled hospital admission of healthcare workers with COVID-19 using Cox regression and conditional logistic regression. Long Nguyen et al. [26] assessed the COVID-19 infection risk among healthcare workers in contrast to the general community by examining the effect of PPE on risk. They also used Cox proportional hazards model to calculate multivariate-adjusted hazard ratios (HRs) of a positive test. However, the major limitations of these models are: 1) the individual-specific characteristics, e.g., occupation [46], type of PPE used, experience level, and exposure duration to COVID-19 patients, are not considered [26, 27], and 2) the simple formalism of the models without time-varying stochastic transmissions oversimplifies the complex contagious mechanism of SARS COV-2.

**3.2.2. Data description.** Data collected from multiple sources, namely COVID-19 transmission databases, health surveys/questionaries, U.S. Department of Labor databases, and cross-sectional study of UK-based healthcare workers, are illustrated in Table 1.

**3.2.3. Model variable selection.** Variables from recent findings of SARS-CoV-2 as introduced in Sub-section 3.2.1, were used to select the features. The validation was performed on three main components: the viral transmission probability model, the individual-level infection risk model, and the population-level risk model. Regarding the viral transmission probability model, we included the following covariates in the model: *Age*, *Cancer*, *Resp*, *Obes*, *Smoker*, *Allied_prof*, *Dental_staff*, *Doctor*, *Pub_trans*, *C_contact*, *AGP*, *PPE_train*, *Lacked_PPE*, *Cont_wo_PPE*, and *Imp_PPE*. These are significant factors suggested by the original cross-sectional study [54]. The description of these variables is summarized in S1 Table. To validate the individual-level infection risk model, the U.S. Department of Labor O*Net database was employed to quantify the risk score for healthcare-related occupations, where virus exposure time and duration and working environment were considered. For the population-level risk model, the PPE sufficiency level, regional infection risk and the hospitalization data of HCP were selected to estimate population-level infection risk in California and Texas medical centers [49, 50] and implement a surrogate method for model validation. The description of these variables is summarized in S1 Table.

**3.2.4. Model validation of viral transmission probability estimation using multivariate logistic regression.** To validate the logistic regression introduced in Sub-section 2.1., we considered different protective and risk factors for COVID-19 in the data set of UK-based healthcare workers [54] and modelled the association between these covariates and the COVID-19 infection status using multivariable logistic regression. The data set provides 6263 responses in

**Table 1. Sources of databases information including source, nation, updated time, and owner.**

| Data source | Nation | Updated time | Owner |
|---|---|---|---|
| Characteristics of HCP with COVID-19 [47] | US | July 16th, 2020 | U.S. CDC |
| COVID-19 transmission dynamics data [48] | Taiwan | Apr 2nd, 2020 | Taiwan CDC |
| California COVID-19 Health Surveys [49] | US | Sep 31st, 2020 | California COVID-19 Health Center |
| Texas Health Center COVID-19 Survey [50] | US | Oct 7th, 2020 | Texas Health Center |
| O*Net database [51] | US | Nov 16th, 2020 | U.S. Department of Labor |
| COVID-NET database [52] | US | Aug 28th, 2020 | U.S. CDC |
| Texas COVID-19 Data [53] | US | Apr 29th, 2021 | Texas Department of State Health Services |
| Cross-sectional observational study of UK-based HCP [54] | UK | May 25th, 2020 | The authors |

**Table 2. Estimated coefficients and their statistical significance for the multivariate logistic regression model.**

| Variables | Description | Coefficient estimates | SE | p-value |
|---|---|---|---|---|
| Intercept | Intercept term | -0.5953 | 0.1497 | 6.98e-05***[a] |
| Age | Age of HCP | -0.0120 | 0.0028 | 1.77e-05*** |
| Cancer | HCP's comorbidities include cancer | 0.5296 | 0.2407 | 0.0277* |
| Resp | HCP's comorbidities include respiratory disease | 0.2020 | 0.0947 | 0.0328* |
| Obes | HCP's comorbidities include obesity | 0.3055 | 0.0872 | 0.0004*** |
| Smoker | HCP is a current smoker or ex-smoker within one year | -0.2490 | 0.1053 | 0.0180* |
| Doctor | HCP is a current smoker or ex-smoker within one year | 0.1514 | 0.0662 | 0.0222* |
| Allied_prof | HCP is a dentist or a dental staff | -0.2282 | 0.0852 | 0.0074** |
| Dental_staff | HCP is a doctor | -0.7018 | 0.2113 | 0.0008*** |
| Pub_trans | HCP uses public transport to travel to work | 0.2728 | 0.0693 | 8.31e-05*** |
| C_contact | Having regular clinical contact with suspected or confirmed COVID-19 patients | 0.2949 | 0.0724 | 4.63e-05*** |
| AGP | Having regular exposure to aerosol generating procedures (AGPs) performed in suspected or confirmed COVID-19 patients | -0.2201 | 0.0663 | 0.0009*** |
| PPE_train | Having sufficient training in PPE use before handling patients | -0.1708 | 0.0666 | 0.0104* |
| Lacked_PPE | Lacked access to PPE items for clinical contact with suspected or confirmed COVID-19 patients | 0.3237 | 0.0776 | 3.03e-05*** |
| cont_wo_PPE | Frequency of contacting without PPE (classified into never, rarely, sometimes, often, and always) | 0.3261 | 0.0768 | 2.21e-05*** |
| Imp_PPE | HCP has used improvised (customized) PPE | -0.2070 | 0.0865 | 0.0166* |

[a] Significance codes: $p \approx 0$ '***', $p < 0.001$ '**', $p < 0.01$ '*', AIC: 7317.7

which a composite outcome was present in 1,806 (29.4%) HCP, of whom 49 (0.8%) HCP were admitted to hospitals, 459 (7.5%) were tested positive for SARS-CoV-2, and 1,776 (28.9%) HCP were self-isolated. The covariates included in the model were reported in Sub-section 3.2.3. The estimated coefficients with their standard errors (SEs) and their statistical significance indicated by p-value are shown in Table 2.

According the table, the most significant variables (p-value < 0.001) that influence the disease transmission probability are *Age*, *Obes*, *Allied_prof*, *Dental_staff*, *Pub_trans*, *C_contact*, *AGP*, *Lacked_PPE*, and *cont_wo_PPE*. The model goodness-of-fit was further assessed by the Akaike information criterion (AIC) and 10-fold cross validation. The AIC value for the above model was 7317.70 and that for the null model was 7449.75. The 10-fold cross validation accuracy was calculated to be 78.23%, which showed that the performance on test data was relatively good.

**3.2.5. Model validation of the individual-level infection risk.** To validate to infection risk model at the individual level, six occupations were considered using the U.S. Department of Labor O*Net database [51]. We also introduced a new variable called occupational-specific risk score denoted as *ORS* to account for the differences in infection risk among different occupations. The score was computed as:

$$ORS = \frac{(CO + PP + EI)}{3\phi} \times \frac{N_{hours}}{\max\{N_{hours}\}} \tag{11}$$

where $\max\{N_{hours}\}$ is the maximum working hours per week of 6 occupations, and $\phi$ is the scaling parameter. The description of those variables $CO$, $PP$, $EI$, and $N_{hours}$ are summarized in S1 Table. Because of the limited longitudinal data, our strategy was to validate the individual infection risk model using hypothesized scenarios of different occupational settings.

Particularly, we made four main assumptions: 1) the individual-risk is the same for every individual working under the same conditions (e.g., same occupation), 2) all patients are confirmed cases, i.e., there is only one compartment $IC$, 3) the probabilities of viral transmission from all patients are the same for each occupation, and 4) the probability of viral transmission estimate for confirmed infectious patients, denoted as $\widehat{p}_{IC}^{(t_1:t_2)}$, is equal to $ORS/\max\{ORS\}$, where $\max\{ORS\}$ is the maximum $ORS$ score among 6 occupations, which guarantees $0 \leq p_{IC}^{(t_1:t_2)} \leq 1$. This is the surrogate approach for approximating the transmission probability defined in Eq (5) in the scenario of limited individual-level data. Consequently, Eq (3) is reduced to:

$$PIR_{i,j}^{(t_1:t_2)} = \sum_{m=1}^{|C^{(\cdot)}|} (1 - \widehat{p}_{IC}^{(t_1:t_2)})^{m-1} \widehat{p}_{IC}^{(t_1:t_2)} \tag{12}$$

Lastly, the total number of contacts $|C^{(\cdot)}|$ was fixed to be 5 and the value $\phi$ was set to 20. Next, the risk was estimated using Eq (12), and the results are summarized in Table 3.

The results of the individual-level model indicated a strong positive association between the estimated risk $PIR_{i,j}^{(\cdot)}$ and virus transmission probability $p_{IC}^{(\cdot)}$, in which the top three occupations that have the highest risk were registered nurses, medical assistants, and respiratory therapists. Their associated $PIR_{i,j}^{(\cdot)}$ values were 0.2262, 0.2119, and 0.1575 respectively, which were relatively high when $|C^{(\cdot)}| = 5$.

**3.2.6. Model validation of the population-level infection risk.** The population-level infection risk was validated based on the total of confirmed COVID-19 cases of HCP reported to the CDC. The number of positive COVID-19 cases of HCP in the US up to April 9, 2020, is presented in Fig 4.

According to Fig 4, there was a strong association between the number of positive cases among non-HCP and the number of cases among HCP by date of symptom onset. In addition, the risk of infection among HCP was closely related to the total number of positive tests among HCP and the patient loads that HCP needed to handle. For population-level, we used the following selected features: $SOH_{time}$, $CS$, $PPE_{SL}$, $ORS$. The description of those is elaborated in S1 Table. Based on Eq (10), population-level risk estimation was reduced to a regressive equation with equal weights assigned to each variable as:

$$\widehat{PIR}_i = \frac{1}{4}(E_{SOH_{time}}[PIR_{i,j}^{(\cdot)}] + E_{CS}[PIR_{i,j}^{(\cdot)}] + E_{PPE_{SL}}[PIR_{i,j}^{(\cdot)}] + E_{ORS}[PIR_{i,j}^{(\cdot)}]) \tag{13}$$

where $E_X[PIR_i^{(\cdot)}]$ is the expected value of $PIR_i^{(\cdot)}$ over the distribution of the variable $X$ and *Value*

**Table 3. Estimated individual-level infection risk for six different occupational settings.**

| Occupations | ORS | $\widehat{p}_{IC}^{(t_1:t_2)}$ | $PIR_{i,j}^{(t_1:t_2)}$ |
|---|---|---|---|
| Registered Nurses | 95.67 | 0.05 | 0.2262 |
| Personal Care Aides | 48.54 | 0.0254 | 0.1206 |
| Nursing Assistants | 59.08 | 0.0309 | 0.1451 |
| Medical Assistants | 89 | 0.0465 | 0.2119 |
| Licensed Nurses | 52.94 | 0.0277 | 0.1309 |
| Respiratory Therapists | 64.47 | 0.0337 | 0.1575 |

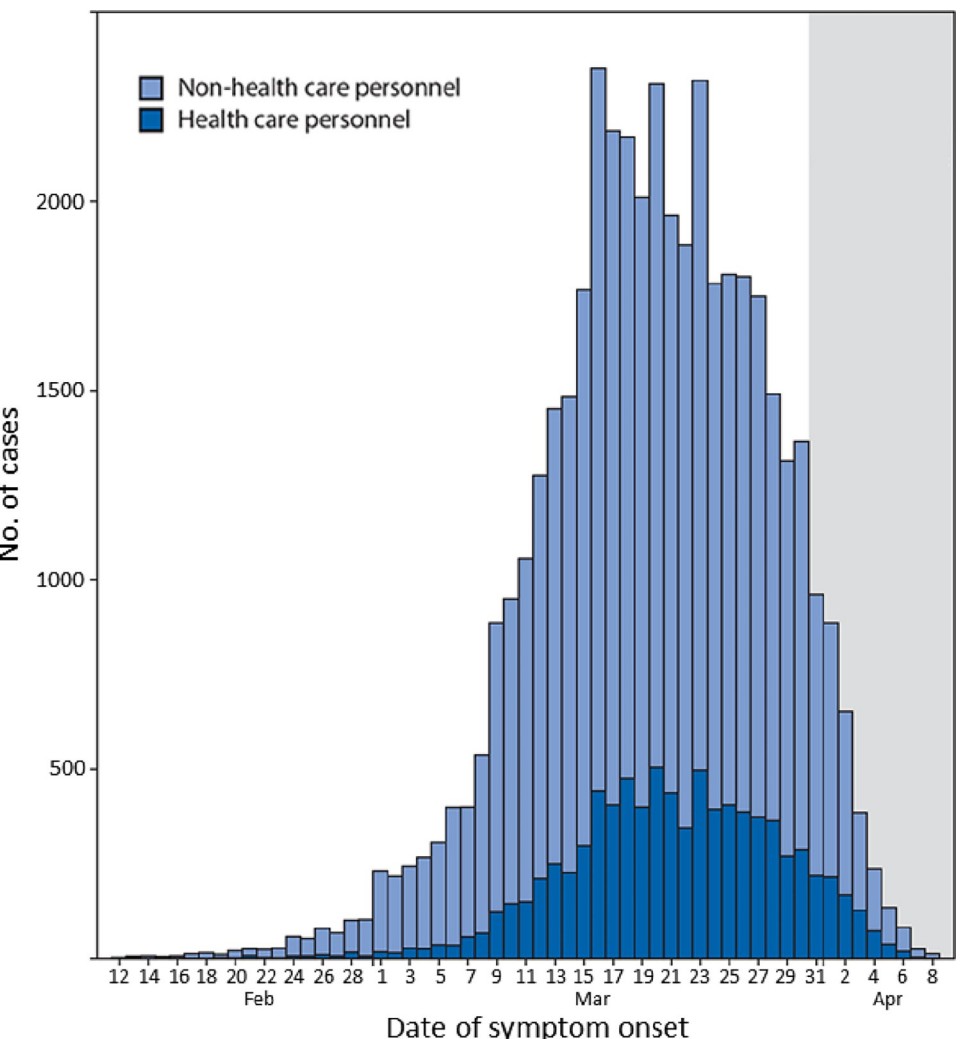

**Fig 4. Daily number of laboratory-confirmed positive COVID-19 cases by date of symptom onset of health care personnel and non-health care personnel (N = 43968) in the US from February 12 to April 9, 2020 [47].**

$(X)$ is the value set of $X$, $E_X[PIR_i^{(\cdot)}]$ is estimated as:

$$E_X[PIR_i^{(\cdot)}] = \sum_{x \in Value(X)} P(X = x)E[PIR_i^{(\cdot)}|X = x] \qquad (14)$$

The population-level infection risk model was validated using the COVID-19 data from health centers in Texas, California and other relevant sources as presented in Sub-section 3.2.2 and Table 1. The accessible HCP COVID-19 data of Texas and California were PPE sufficiency level, the total number of hospitalizations, and the percentage of ICU beds available. So, we assumed the distributions and the expected value of $PER_i^{(\cdot)}$ over the other variables to be the same for both states. The expected values of $PER_i^{(\cdot)}$ was computed using Eq (14) (see Table 4).

In Table 4, $E_{PPE_{SL}}[PIR_i^{(\cdot)}]$ was estimated using the PPE lacking information in health centers in Texas and HCP surveys in California. The value of $E_{ORS}[PIR_i^{(\cdot)}]$ was estimated by averaging

**Table 4. Estimated value and distribution of the selected features used in two case studies to estimate the infection risk in Texas and California.**

| Features | Texas | California |
|---|---|---|
| Time from symptom onset to hospitalization | The distributions of $SOH_{time}$ and $CS$ are estimated from [47, 48]. | |
| | $P(SOH_{time} < 0) = 0.34, P(SOH_{time} \in [0, 3]) = 0.21\ P(SOH_{time} \in [4, 5]) = 0.05, P(SOH_{time} \in [6, 7]) = 0.02,$ | |
| | $P(SOH_{time} \in [8, 9]) = 0.16, P(SOH_{time} > 9) = 0.21\ E_{SOH_{time}}[PIR_i^{(\cdot)}] = 5.99 \times 10^{-3}$ | |
| Clinical severity of patients | $P(CS = "\text{Severe pneumonia}") = 0.01, P(CS = "\text{ARDS/Sepsis}") = 0.01$ | |
| | $P(CS = "\text{Asymptomatic}") = 0.08, E_{CS}[PIR_i^{(\cdot)}] = 3.89 \times 10^{-3}$ | |
| PPE sufficiency level | PPE sufficiency levels were averaged to estimate $E_{PPE_{SL}}[PIR_{i,j}^{(\cdot)}]$ to be 0.0065 | $E_{PPE_{SL}}[PIR_{i,j}^{(\cdot)}]$ was estimated to be 0.744 |
| ORS | $E_{ORS}[PIR_i^{(\cdot)}]$ was estimated to be the average over of $PIR_{i,j}^{(t_1:t_2)}$ over all occupations at 0.0173 | |
| Estimated $\widehat{PIR}_i$ | $\widehat{PIR}_{Texas} = 0.0084$ | $\widehat{PIR}_{California} = 0.0132$ |

the values of $PIR_{i,j}^{(\cdot)}$ over all occupations. The estimated $\widehat{PIR}_i$ values for Texas and California were 0.0084 and 0.0132, respectively.

## 4. Discussion

Hospital-acquired infections of communicable viral diseases are posing a challenge to health-care workers globally. HCP is facing a consistent risk of hospital-acquired infections, and subsequently higher rates of morbidity and mortality. Therefore, mitigating and preventing nosocomial infections in hospitals is an urgent and important task to lower the risk of contracting CVDs for HCP, guarantee adequate availability of PPE and develop well-informed strategies to protect health-care workers from contracting CVDs. In this paper, we have developed a proposed probabilistic model characterizes the dynamics of the disease transmission in HCP over time, in which the domain-knowledge-driven risk analysis framework can quantify both the individual-level and population-level infection risk. We validated the model at both levels using two main approaches, namely the variance-based sensitivity analysis using the simulated data and the COVID-19 case study. The sensitivity analysis indicated that the uncertainty in the HCP infection risk is attributed to 2 variables: the number of close contacts and the viral transmission probability. The COVID-19 case study showed that the occupations with the highest risk are registered nurses, medical assistants, and respiratory therapists. In addition, the results indicated the significant risk and protective factors of the COVID-19 transmission risk of HCP.

In our sensitivity analysis, we focused only on two key variables, namely viral transmission probability and the number of close contacts between HCP and patients. Specifically, the sensitivity of the infection risk to those input variables was measured by the amount of variance caused by changing the inputs. We divided our analysis into two parts: 1) the measure of sensitivity of $PIR_{i,j}^{(\cdot)}$ to $p_{X(m),k(m) \to j}^{(\cdot)}$ and close contact sequence, and 2) response surface of the mean and variance of $PIR_{i,j}^{(t_1:t_2)}$ to $|C^{(\cdot)}|$ and $p_{X(m),k(m) \to j}^{(\cdot)}$. The results of the sensitivity analysis revealed that the output $PIR_{i,j}^{(\cdot)}$ will be significantly increased when the viral transmission probability $p_{X(m),k(m) \to j}^{(\cdot)}$ and the number of contacts increases. In addition, the results in the second part indicated that $E[PIR_{i,j}^{(\cdot)}]$ quickly converged to one as $|C^{(\cdot)}| \to \infty$, and the convergence rate was faster if $P_{low}$ took higher values. Based on the response surface of $Var[PIR_{i,j}^{(\cdot)}]$, higher values of $P_{low}$ and $|C^{(\cdot)}|$ will lead to a lower value of $Var[PIR_{i,j}^{(\cdot)}]$; however, the effect of $P_{low}$ is more significant than that of $|C^{(\cdot)}|$. The value of $Var[PIR_{i,j}^{(\cdot)}] \to 0$ as $|C^{(\cdot)}| \to \infty$ and dropped to nearly 0 after only four close contacts when $P_{low} = 0.5$.

After performing the sensitivity analysis, the logistic regression for estimating viral transmission probability $\widehat{p}_{X,r\rightarrow j}^{(t_1:t_2)}$ was validated using the cross-sectional observational study of UK-based healthcare workers. Based on the coefficient estimates of the variables in the built multivariate logistic regression model, *Age*, *Smoker*, *Allied_prof*, *Dental_staff*, *AGP*, *PPE_train*, *Imp_PPE* were the protective factors, whereas the risk factors were *Cancer*, *Resp*, *Obes*, *Doctor*, *Lacked_PPE*, *cont_wo_PPE*, *Pub_trans*, *C_contact*. Surprisingly, advanced age, being a smoker or ex-smoker within one year, and having regular exposure to aerosol-generating procedures performed on COVID-19 patients decreased the infection risk. This result seems counter-intuitive at first, but they are confounders because it was shown that HCP working directly with suspected or confirmed COVID-19 patients tended to be more cautious and self-aware in clinical environments [55]. Therefore, they had sufficient self-protection and took containment measures; however, healthcare workers in non-communicable viral disease departments, who were potentially exposed to contagious viruses, did not have sufficient training on how to use PPE and deal with infectious diseases and lack of access to PPE and isolation equipment [56]. However, the model has several limitations. First, because we did not have access to information on HCP contact with patients and coworkers, we assumed the estimated viral transmission probability as a measure averaged over all individuals. Second, the data were gathered using surveys and questionnaires, which are subject to selection and recall bias. Third, the use of a composite outcome (including HCP with COVID-19 symptoms, HCP being exposed to risk factors, and lab-confirmed HCP infections) may have resulted in overestimation or underestimation of the infection risk.

We validated the individual-level infection risk model, implemented the model using the two-parameter regressive equation, and estimated the individual risk for six occupations. The results highly depend on the pre-defined parameters, which can be estimated in healthcare settings when data are available. It was shown that healthcare workers and nurses are frequently in close contact with COVID-19 patients, which therefore increases the risk for acquiring SARS-CoV-2 virus [57]. Because HCP can acquire infection through various pathways apart from direct patient care, such as exposure to colleagues, family members, or people in the community, the time-varying risk estimation in the model can provide informed decisions for screening HCP for COVID-19 before workplace entry. The individual risk model can be improved and more specific to better model the transmission dynamics, e.g., a model that incorporates the quantification of indoor airborne infection risks using a probabilistic framework [58]. In addition, we do not assume that the recovered patients confer immunity to reinfection when being released from isolation This statement can be further clarified that even the patients are fully recovered after getting contracted with COVID-19 (or any communicable viral diseases in general) and being released back to the population, they are still under the risk of reinfection with the same disease strain or other strains. However, reinfection with the same strain is very rare. Hence, if the HCP were recovered from the disease, they might get infected again; however, they still confer some degree of immunity from subsequent infection. Therefore, we can consider adding a new group of patients in our model called "reinfected patients". Moreover, the same idea can be applied for the vaccinated population, in which people have vaccine-induced immunity.

For model validation at the population level, we considered two case studies to estimate the risk of infection of HCP in Texas and California states. Both states have a high number of lab-confirmed SARS-CoV-2 patients. The average number of hospitalizations in Texas and California were 16843 cases/day and 4219 cases/day, respectively. However, the infection risk in Texas was 0.0084 which was lower than the risk in California (0.0132). This was mainly due to the difference in patient load for each HCP per day and the two states' PPE sufficiency level.

From Table 4, the average PPE sufficiency level in California was only 0.744 as opposed to 0.9355 in Texas, and the average percentage of ICU beds available per 100,100 people in Texas was significantly higher than that in California, which implies heavier patient loads in California. The model also made some important assumptions: 1) close contacts with COVID-19 patients are independent and there is no viral transmission among HCP, and 2) protective/risk factors are well-defined and sufficient to estimate the risk of infection.

## 5. Conclusion and future work

The paper proposed a time-variant infection risk analysis model to characterize the dynamic of the disease infection risk in HCP over time and a domain-knowledge-driven infection risk to quantify the complexities of HCP's risk of CVDs in healthcare settings. The infection risk analysis model for HCP was estimated at both individual and population levels. The individual-level risk model was built based on the population grouping concept of the well-established epidemiological SEIR model with the consideration of the time-varying confounders to capture the dynamical contagious disease transmission mechanism. At the population-level, three subsets of features were constructed and represented by a Bayesian network, from which the probability of viral transmission from patients to HCP was estimated. To validate our methods, we have incorporated the data from multiple data sources from the US, the UK, and Taiwan for the COVID-19 case study, which contains the information about potential factors that affect COVID-19 transmission mechanism; and the domain knowledge of similar contagious diseases such as SARS or MERS from the relevant studies to estimate the risk of COVID-19 infection of HCP. For individual-level risk estimation, the model was founded on the SEIR compartmental model and developed for the occupational-specific and individualized infection risk model. As a result, the model can accurately capture the infection risk varying over time under the control of those individual time-varying confounders, and it is also able to account for the intrinsic stochastic transmission mechanisms. At the population level, the Bayesian network formalism can accommodate the limited data scenario, and it can update the parameters when more data are available. The results from two case studies are interpretable at the population level, which showed infection risk in California is higher than in Texas because of the heavier patient loadings and shortage of PPE. The major limitations of the CDC's interim guideline for risk assessment, which is inadequate in quantifying the risk of infection in an individualized HCP, have been addressed by our model. The model would significantly endorse the PPE allocation and safety plans for HCP and enhance the crisis-level staffing strategies in facilities with the staffing shortages. Longitudinal experimental designs are required to collect more COVID-19 data among HCP to validate the proposed model properly. Future work would involve: 1) model assumption validation when more data are available and sufficient, 2) model modification and reformulation if the assumptions are violated (e.g., independence assumption and new vaccinated or "reinfected" population), and 3) validating the model with the other related case studies of communicable viral diseases.

### Summary table

What was already known on the topic

- Hospital-acquired infections of communicable viral diseases are posing a challenge to healthcare workers globally.

- Healthcare personnel (HCP) is facing a consistent risk of hospital-acquired infections, and subsequently higher rates of morbidity and mortality.

- Mitigating and preventing nosocomial infections in hospitals is an urgent and important task to lower the risk of contracting CVDs for HCP.

- Previous mathematical models and statistical methods to model the nosocomial infection risk in HCP fail to capture the time-dependent disease transmission processes and the effects of individual risk and protective factors.

- What this study added to our knowledge

- Our proposed probabilistic model characterizes the dynamics of the disease transmission in HCP over time

- The knowledge-driven risk analysis framework can quantify both the individual-level and population-level infection risk.

- The sensitivity analyses indicated that the uncertainty in the HCP infection risk is attributed to 2 variables: the number of close contacts and the viral transmission probability.

- The COVID-19 case study showed that the occupations with the highest risk are registered nurses, medical assistants, and respiratory therapists.

- The results indicated that age, smoking status, occupation, PPE use, using public transport, close contacts with patients, and having regular exposure to aerosol-generating procedures are significant factors.

## Supporting information

**S1 Table. Characteristics of the selected features and their associated databases.**
(DOCX)

**S1 File. The expectation and variance of the potential individual risk.**
(DOCX)

## Author Contributions

**Conceptualization:** Phat K. Huynh, Arveity R. Setty, Quan M. Tran, Om P. Yadav, Nita Yodo, Trung Q. Le.

**Data curation:** Phat K. Huynh.

**Formal analysis:** Phat K. Huynh.

**Funding acquisition:** Trung Q. Le.

**Investigation:** Phat K. Huynh, Arveity R. Setty, Trung Q. Le.

**Methodology:** Phat K. Huynh, Quan M. Tran, Om P. Yadav, Trung Q. Le.

**Software:** Phat K. Huynh.

**Supervision:** Phat K. Huynh, Trung Q. Le.

**Validation:** Phat K. Huynh, Arveity R. Setty, Om P. Yadav, Trung Q. Le.

**Visualization:** Phat K. Huynh, Arveity R. Setty, Trung Q. Le.

**Writing – original draft:** Phat K. Huynh.

**Writing – review & editing:** Phat K. Huynh, Arveity R. Setty, Quan M. Tran, Om P. Yadav, Nita Yodo, Trung Q. Le.

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
