## [Decision Letter · Decision Letter 0]

7 Apr 2022

PONE-D-22-00970A Domain-knowledge Modeling of Hospital-acquired Infection Risk in Healthcare Personnel from Retrospective Observational Data: A Case Study for COVID-19PLOS ONE

Dear Dr. Le,

Thank you for submitting your manuscript to PLOS ONE. After careful consideration, we feel that it has merit but does not fully meet PLOS ONE’s publication criteria as it currently stands. Therefore, we invite you to submit a revised version of the manuscript that addresses the points raised during the review process.This is a relevant and well written manuscript. Please make sure that your assumptions correspond to the worst case scenario, as suggested by reviewer 2, and justify (or change) your data sharing policy. Also, please reply to all aspects raised by reviewers 2 and 3.

We look forward to receiving your revised manuscript.

Kind regards,

Anete Trajman

Academic Editor

PLOS ONE

Journal Requirements:

Reviewers' comments:

Reviewer's Responses to Questions

**Comments to the Author**

1. Is the manuscript technically sound, and do the data support the conclusions?

Reviewer #1: Yes

Reviewer #2: Yes

Reviewer #3: Yes

2. Has the statistical analysis been performed appropriately and rigorously? 

Reviewer #1: Yes

Reviewer #2: Yes

Reviewer #3: Yes

3. Have the authors made all data underlying the findings in their manuscript fully available?

Reviewer #1: Yes

Reviewer #2: No

Reviewer #3: No

4. Is the manuscript presented in an intelligible fashion and written in standard English?

Reviewer #1: Yes

Reviewer #2: Yes

Reviewer #3: Yes

5. Review Comments to the Author

Reviewer #1: The study successfully investigates the infection risk on a bayesian network model and the model is validated against U.S. Department of Labor O*Net database. Public health experts, engineers, and epidemiologists can use the modified model when selecting different measures to reduce the infection risk from various respiratory diseases indoors, allowing informed decisions concerning environmental control.

Reviewer #2: First of all, I must congratulate you for the interesting and well presented work.

The problem is highly relevant to society, and the motivation and intended impacts are well introduced. Statements are well supported by the bibliography.

Methodology is also sound and I agree with the statement of the two main contributions of the work ("a novel probabilistic model characterizes the dynamics of the disease transmission in HCP over time and a domain-knowledge risk analysis model that quantifies both the individual-level and population-level infection risk").

However, I am mostly concerned about two modeling assumptions. First, that infection confer no immunity to the HCP and second, that I don't see clearly in the model the removal of infected HCP (confirmed or suspected) from regular activities. More specifically, I think that these assumptions may lead to an overestimation of cases which may affect parameter estimation. Nonetheless, these assumptions may still be valid in considering the worst-case limiting scenario which is still an interesting and valid first approach. I understand that changing these assumptions would be best left to future works, but I still believe that the limitations imposed by these assumptions should be more carefully addressed in the revision.

I also recommend the improvement of data availability. If data can be provided upon request, I do not understand why it cannot be made publicly available in an online repository such as github or gitlab. If not possible, please justify your decision.

Tables and figures are of sufficient quality, but their readability could be improved by reducing the use of abbreviations a little. Most importantly, I request that you improve the naming of covariates in the model, exposed in section 3.2.4 and Table 2. Resolution of figures should be improved for final manuscript.

Additionally, there are a couple of incorrect closures of parenthesis in the last paragraph of section 2; please correct these and verify carefully if such mistakes escaped our eyes elsewhere. It would be appreciated if line numbering was incorporated in the revision to facilitate the correction of such small problems for the final manuscript.

Reviewer #3: The submitted manuscript entitled "A Domain-knowledge Modeling of Hospital-acquired Infection Risk in Healthcare Personnel from Retrospective Observational Data: A Case Study for COVID-19" by Huynh et al. aims to develop infection risk models at both individual and population levels. The authors explore traditional SEIR dynamics to build an individual risk model and a Bayesian framework for the population indicator. The scope of the paper is well defined. I have only a shortlist of minor revisions that I would like to see addressed in a revised manuscript version.

Suggested revisions

- The abstract lacks a definition of “PPE’’

- The abstract section results are populated with numbers without any explanation or context. It is hard to tell if the results are good or bad from the numbers only. Please refine the abstract to reflect the main takeaway points for a broader audience.

- Introduction: “there has been an increasing hospital outbreaks” → there has been an increasing number of hospital outbreaks.

- Introduction: “Quantitative models have been used as an alternative to mathematical models.” I have the impression that Quantitative and mathematical models are synonyms. Perhaps the authors could use the alternative form: Other quantitative approaches have been used…

- Introduction: “Section 2 discusses about the model formulation and model validation”

- Introduction: I missed a review of modeling studies for nosocomial HCP infections at both individual and population levels. Since the paper is devoted to those two different problems, that could be reflected in the introduction. Instead, the authors briefly describe some previous work without that explicit distinction.

- Introduction: To overcome the above research gaps, this paper proposes a probabilistic domain-knowledge model

The term ``domain-knowledge’’ appears throughout the manuscript. I wonder if there is a precise definition for the term in the context of this study (or references) that could be included in the introduction.

-Materials and Methods: “…(1) an individual-level infection risk model that quantifies the risk of infection of an HCP… (2) a population-level infection risk indicator model that estimates the infection risk under working conditions at a medical facility”

Odd sentence construction. I suggest: “(1) an individual-level infection risk model for HCP and (2) a population-level model that estimates the infection risk under working conditions… ”

- Materials and methods, Section 2.1: In Eq. 4, the first equation is substituted by a summation indexed by m, and new variables h_m appear. Perhaps this is a standard calculation and I am missing some elementary steps, but it was unclear how the breakdown of h into h_m is done. Please clarify.

- Materials and methods, Section 2.1: “where is the length of the close contact with person ()”

It seems it should be m instead of r

- Materials and methods, Section 2.1:

The order in which the individual model is presented seems counterintuitive. After reading it carefully, I understood that the first step would be to calculate the probabilities P(),()→j from the logistic regression with the covariate data, and then the risk indicator could be estimated. The authors leave the logistic regression as the last step, which is confusing. Please consider addressing this point to improve the text clarity.

- Materials and Methods, Section 2.2: “We denote (∙) as the abbreviated notation for the function of , (∙) and in Eq. (9)”

It is unclear the role of Eq 9 in this sentence and the authors should avoid mentioning an equation before it appears in the text. Please clarify.

- Materials and Methods, Section 2.2: “the population risk (∙) is estimated using a Bayesian network…”

The variable (∙) has not been defined. I assume the authors meant (∙) instead

- Results: Caption of Fig. 3 “Three data tips at = {0.05, 0.2, 0.5} were”

It seems an incomplete sentence.

- Results: Sections 3.2.1 are and 3.2.2 are devoted to well-known contributing factors and related work on HCP infection risk for COVID-19. Both sections do not contain results and should be part of the introduction or discussion.

- Results. Section 3.2.6: In Eq. 10, what is the value of the scaling parameter Φ?

- Results. Section 3.2.6: If ^ (1:2) is / max{}, then it is unclear why Eq 6 was mentioned in the Materials and Methods section. Please clarify or omit any equation that is not used in the results section.

- Discussion: It is good practice to introduce a first paragraph reviewing the main problem, what methods were developed, and the main takeaways of a study. Please consider adding such a paragraph.

6. PLOS authors have the option to publish the peer review history of their article (what does this mean?). If published, this will include your full peer review and any attached files.

Reviewer #1: No

Reviewer #2: No

Reviewer #3: No

---

## [Author Response · Author response to Decision Letter 0]

19 Jun 2022

The response has been provided in the attached file: PLOSONE_Reviewer_Response_June16.docx

---

## [Decision Letter · Decision Letter 1]

29 Jul 2022

A Domain-knowledge Modeling of Hospital-acquired Infection Risk in Healthcare Personnel from Retrospective Observational Data: A Case Study for COVID-19

PONE-D-22-00970R1

Dear Dr. Le,

We’re pleased to inform you that your manuscript has been judged scientifically suitable for publication and will be formally accepted for publication once it meets all outstanding technical requirements.

Kind regards,

Anete Trajman

Academic Editor

PLOS ONE

Additional Editor Comments (optional):

Reviewers' comments:

Reviewer's Responses to Questions

**Comments to the Author**

1. If the authors have adequately addressed your comments raised in a previous round of review and you feel that this manuscript is now acceptable for publication, you may indicate that here to bypass the “Comments to the Author” section, enter your conflict of interest statement in the “Confidential to Editor” section, and submit your "Accept" recommendation.

Reviewer #2: All comments have been addressed

Reviewer #3: All comments have been addressed

2. Is the manuscript technically sound, and do the data support the conclusions?

Reviewer #2: Yes

Reviewer #3: (No Response)

3. Has the statistical analysis been performed appropriately and rigorously? 

Reviewer #2: Yes

Reviewer #3: (No Response)

4. Have the authors made all data underlying the findings in their manuscript fully available?

Reviewer #2: Yes

Reviewer #3: (No Response)

5. Is the manuscript presented in an intelligible fashion and written in standard English?

Reviewer #2: Yes

Reviewer #3: Yes

6. Review Comments to the Author

Reviewer #2: Thank you for addressing our concerns. The manuscript is more polished now and my main questions were sufficiently addressed.

Reviewer #3: (No Response)

7. PLOS authors have the option to publish the peer review history of their article (what does this mean?). If published, this will include your full peer review and any attached files.

Reviewer #2: No

Reviewer #3: No

---

## [Editor Report · Acceptance letter]

9 Nov 2022

PONE-D-22-00970R1 

A Domain-knowledge Modeling of Hospital-acquired Infection Risk in Healthcare Personnel from Retrospective Observational Data: A Case Study for COVID-19 

Dear Dr. Le:

I'm pleased to inform you that your manuscript has been deemed suitable for publication in PLOS ONE. Congratulations! Your manuscript is now with our production department. 

Kind regards, 

on behalf of

Professor Anete Trajman 

Academic Editor

PLOS ONE